# Children's exploratory play tracks the discriminability of hypotheses

Max H. Siegel 1,2✉, Rachel W. Magid1,2, Madeline Pelz1, Joshua B. Tenenbaum1 & Laura E. Schulz 1✉

Effective curiosity-driven learning requires recognizing that the value of evidence for testing hypotheses depends on what other hypotheses are under consideration. Do we intuitively represent the discriminability of hypotheses? Here we show children alternative hypotheses for the contents of a box and then shake the box (or allow children to shake it themselves) so they can hear the sound of the contents. We find that children are able to compare the evidence they hear with imagined evidence they do not hear but might have heard under alternative hypotheses. Children (N = 160; mean: 5 years and 4 months) prefer easier discriminations (Experiments 1-3) and explore longer given harder ones (Experiments 4-7). Across 16 contrasts, children's exploration time quantitatively tracks the discriminability of heard evidence from an unheard alternative. The results are consistent with the idea that children have an "intuitive psychophysics": children represent their own perceptual abilities and explore longer when hypotheses are harder to distinguish.

---

[1] Massachusetts Institute of Technology, Cambridge, MA, USA. [2]These authors contributed equally: Max H. Siegel and Rachel W. Magid. ✉email: maxs@mit.edu; lschulz@mit.edu

Young children are remarkable learners, constructing intuitive theories that support prediction, explanation, intervention, and discovery. These early-emerging abilities arguably lay the foundation for scientific inquiry[1,2]. However, both scientific inquiry and everyday learning are difficult in part because we can often get only indirect evidence to test our hypotheses: we want to know the composition of stars but can only measure the light they emit and absorb; we want to understand the neural basis of cognition but can only observe changes in blood flow. In science, we bridge the gap between ordinary perception and the otherwise unobservable and unknown through extensive causal chains. In everyday life, we do not use fancy telescopes or imaging equipment but must bridge an analogous gap: we hear a crash in another room and infer that something heavy was dropped; we see a curtain move and infer the cat behind it. These are ordinary, common-sense inferences—ones even a child might make—but they depend on an extraordinary capacity: the ability to use our understanding of the physical world to reason back from what we perceive to its probable unobserved causes.

We focus on a paradigmatic case of everyday exploration: trying to figure out what is inside a box by shaking it. Most of us have shaken a wrapped present at some point to try to guess its contents, suggesting that we think we can imagine how different items would sound given the motion of the box. Consistent with this intuition, studies suggest that adults, and even infants[3–5], can mentally simulate the physical interactions of moving objects on short timescales. Such simulations might help us guess what is in a box, but they might also let us estimate the relative discriminability of different hypotheses and thereby make critical decisions about how to explore (e.g., how long to shake the box, how hard to shake it, or which of multiple boxes might be most worth shaking). As in science, a rational learner should be able to estimate the sensitivity of her measurement apparatus (in this case, her perceptual system) to decide what would count as an informative experiment and amount of data, given the alternative hypotheses she is trying to discriminate among[6–9]. Here we ask whether such an "intuitive psychophysics" guides children's exploration. Can children use their intuitive understanding of both the physical world and their own ability to make perceptual discriminations to engage in effective exploration? Do they compare the perceptual evidence they observe with the evidence they think they would have observed under different competing hypotheses?

Our proposal builds on three more basic capacities that we already know children possess: aspects of intuitive physics (i.e., the ability to represent the physical interactions among objects) and intuitive psychology (i.e., the ability to represent the relationship between seeing and knowing), and an ability to make psychophysical discriminations themselves (i.e., to hear the difference between two quite different sounds more easily than the difference between two similar ones). In asking whether children have an "intuitive psychophysics", we are asking whether children can use these abilities to judge whether they themselves will be able to distinguish evidence for different physical interactions. Can children simulate the interactions among physical events and the perceptual consequences of these interactions with sufficient granularity to represent their own ability to discriminate among events? Note that having an intuitive psychophysics need not imply that children can explicitly explain or justify their own judgments (any more than having an intuitive physics requires that children be able to explain their own reasoning about objects and forces). However, to the degree that children have an intuitive psychophysics, they should be able to represent the relative difficulty of discriminating perceptual evidence and these representations should guide their judgment and exploration.

Our study connects to a growing literature in cognitive science, cognitive neuroscience, and AI investigating rational curiosity: learners' tendency to explore more when the expected information gain is higher[10–17]. Classic[18] and contemporary[19,20] work has examined the extent to which adult learning and exploration can be considered to be rational, and developmental studies suggest that even young children explore more when evidence is surprising[21–27] or confounded[28–30]. However, such studies have provided children with perceptually unambiguous evidence and, with the exception of work showing a U-shaped relationship between infant looking-time and the predictability of events[31,32] (see also ref. [5]), looked only at qualitative relationships between children's uncertainty and exploration. In particular, previous studies looking at children's sensitivity to their own uncertainty have considered cases where evidence is surprising (e.g., refs. [25,31]), uninformative with respect to competing hypotheses (e.g., ref. [29]), or cases where children simply do not know the answer to a query (e.g., refs. [16,33,34]). In contrast, here we look at cases where evidence to distinguish hypotheses is available and, in principle, informative, and we ask whether children represent their own ability to make distinctions among the available evidence. Specifically, rather than asking whether children can distinguish two different observations (as one might in a psychophysics experiment), we allow children to observe only one kind of event and we ask whether they recognize that the observation is more discriminable from some hypotheses than others. That is, we are interested in whether children can simulate the evidence they might get under alternative hypotheses and compare the discriminability of observed evidence with unobserved alternatives. Finally, we ask whether there is a precise quantitative relationship between the discriminability of competing hypotheses and children's active exploration.

We report two series of experiments probing children's intuitive psychophysics, considering first children's reasoning about exploration, and second, their decisions about how long to explore. In Experiments 1–3, the experimenter shakes two boxes, generating identical sounds (see Fig. 1). Children are asked to decide which box they want to open to find a target. The only difference between the boxes is the alternative item that might have been in the box and the degree to which it would have been distinguishable from the target based on the sounds. In Experiments 4–7, children get to shake the box themselves to guess which of two alternatives are inside. The alternatives differ only in numerical quantity (e.g., three marbles or six marbles), which we vary across trials, systematically manipulating the discriminability of the hypotheses. Children are allowed to shake the box for as long as they want, allowing us to investigate the extent to which children's free exploration tracks the quantitative discriminability of the alternative hypotheses. In Experiments 1–3, we focus on 4- and 5-year-olds, consistent with previous work on children's active exploration[21–24,28,30,35]. In Experiments 4–7, where we look at children's response to graded numerosity contrasts, we expand the range to 4- to 8-year-olds given the possibility that developmental changes in children's number representations across this age range[36,37] might impact their exploration. Throughout, we adopt the convention in developmental psychology of reporting children's ages as years; months (e.g., mean age of 4 years and 4 months is written 4;4).

## Results
**Experiment 1**. Preliminary studies (see SI) established that children could guess which of two boxes contained a target when the boxes generated two very different sounds when shaken: 100% of children distinguished a soft bean bag from a hard ball, and 100% distinguished eight marbles from two marbles. To establish that

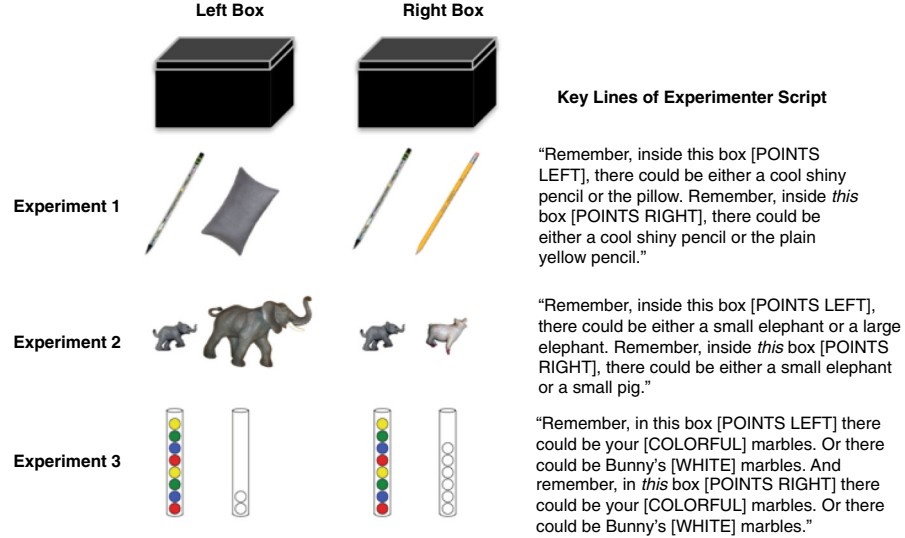

**Fig. 1 Schematic of Experiments 1–3 showing the more discriminable pair on the left and the less discriminable pair on the right (actual order counterbalanced).** The leftmost item in each pair was the target. Only **one** item in each pair (the target) was placed in each box. Because the target was always placed in both boxes, the two boxes in each experiment made the same sound when shaken.

children engage in a relatively rich mental simulation of the physics of the event, we also showed that children were able to distinguish two from eight marbles even when the eight-marble box contained a cloth, muffling the sound ($N = 15$; mean age: 4;4; 86.7% correct; 95% CI [0.67−1], Bernoulli normal approximation CI) and even when the experimenter shook the two-marble box but tilted the eight-marble box back and forth, rather than shaking it ($N = 15$; mean age: 4;11; 86.7% correct; 95% CI [0.67−1]).

Having established that children's intuitive physics can support inferences about the hidden causes of auditory stimuli, we turned to the question of whether children could determine the extent to which perceptual cues are and are not informative, given different competing hypotheses about their latent causes. In Experiments 1 and 2, we looked at participants' inferences when the content of the boxes differed in kind; in Experiment 3, we looked at children's inferences when the contents differed in quantity.

In Experiment 1 (see Fig. 1 and SI for details), children were introduced to two boxes. A pair of objects were placed in front of each box. Each pair consisted of an exciting target object (a pencil with a shiny holographic coating) and a boring distractor. The target was identical in both pairs. In the less discriminable pair, the distractor was an object that would make a very similar sound when shaken inside the box (a standard No. 2 pencil). In the more discriminable pair, the distractor was an object that would make a very different sound when shaken inside the box (a small pillow). The experimenter pointed to the shiny pencil and the boring pencil and told the child, "I'm going to take just one object -- either the shiny pencil or the plain pencil -- and put it in this box here." Then she pointed to the other pair and the other box and said, "And then I'm going to take just one object -- either the shiny pencil or the cotton pillow -- and put it in this box here." She put up an opaque screen and removed all the objects from the child's line of sight. She silently put a shiny pencil in each box and then returned the boxes to the table. She told the child, "Remember, inside this box, there could be either a cool shiny pencil or the plain yellow pencil"; "Remember, inside this box, there could be either a cool shiny pencil or the pillow"; (order and L/R position counterbalanced). The experimenter shook each box generating identical sounds. Children were asked which box they wanted to open to find the target. The

experimenter was not blind to the contents of the box, so to avoid her influencing the child's choice, the left/right positions of the box were fixed and the experimenter looked directly at the child during the prompt. Children ($N = 16$, mean age: 4;7) successfully chose the box where the unheard alternative, the pillow, would have been easier to discriminate from the target (81.2%; 95% CI [0.63−1]).

**Experiment 2**. In Experiment 2, we replicated the design of Experiment 1, except that the more discriminable pair consisted of a small and large plastic elephant; the less discriminable pair consisted of a small plastic elephant and a small plastic pig. Children were told that the baby elephants had been separated from their friends (other plastic elephants housed in a separate container) and were asked to find them. The small elephant was hidden in both boxes. As in Experiment 1, children ($N = 24$; mean age: 4;8) successfully chose the box where the target would be easier to discriminate from the unheard alternative (the large elephant) (79%; 95% CI [0.63−0.96]). Importantly, this is not because children thought this pair was more dissimilar overall; a separate group of children ($N = 24$; mean age: 4;8) asked only which pair was more similar (without a box-shaking task) thought the small elephant and small pig were more dissimilar than the small and large elephant (83%; 95% CI [0.67−0.96]).

**Experiment 3**. In Experiment 3, preregistered on the Open Science Framework (https://osf.io/ytvse/?view_only=abe4554f3ace 483490953768b58efbfc), we looked at whether children could infer the more discriminable of two boxes when the contents differed only in quantity. The less discriminable pair consisted of 8 marbles and 6 marbles; the more discriminable pair consisted of 8 marbles and 2 marbles. Both boxes in fact contained 8 marbles. Children ($N = 24$; mean: 5;0), successfully chose the box associated with the more discriminable (8 vs. 2) pair (75%; 95% CI [0.58−0.92]).

The results of Experiments 1–3 suggest that 4- and 5-year-old children represent the relative discriminability of perceptual evidence. Critically, children's choices were guided not by the evidence they observed (which was identical between choices) but

by its contrast with the unheard alternatives, consistent with the idea that children can simulate novel physical interactions and the perceptual data that will result[3]. Children's ability to represent their own ability to make these perceptual discriminations is consistent with emerging evidence for metacognitive monitoring in young children (see ref. [38] for review) and also suggests that, at least in simple, forced-choice contexts, children can exercise metacognitive control for effective decision-making[39–43].

**Experiments 4–7.** In Experiments 4–7, we looked to see if children's exploration times quantitatively tracked the discriminability of hypotheses. Because we wanted to test children on a range of discriminability contrasts (and because pilot work suggested it was impractical to test children on more than four contrasts at a time), we ran four separate experiments consisting of four contrasts each. The experiments differed only in the contrasts presented. The design and quantitative predictions for the last experiment (Experiment 7), as well as the overall analysis across all 16 contrasts, were preregistered (https://osf.io/dxguw/?view_only=ba3ca1c5ff9346c0a39e731291aa5d5f). See SI for details throughout.

The experimenter introduced two tubes of marbles; each tube contained a different number of marbles, varying in numerosity between one and nine (Fig. 2). Out of the children's sight, the contents of one of the tubes were placed in the box. Children were allowed to shake the box for as long as they liked to try to guess its contents. After each trial, a new pair of tubes were introduced. Children were not given any feedback between trials.

Exploration time was coded from video by a human coder blind to contrast and, independently, by a motion sensor in the box (see SI). The experimenter was not blind to the contents of the box, but was blind to the precise predictions across all sixteen contrasts. The experimenter was positioned alongside the child, out of the child's direct line of sight, and did not interact with the child or the box during the exploration period. The behavioral coding included the time from the moment the child first contacted the box until she identified the contents of the box on each trial. The motion sensor coded the time from the initial motion to the final motion on each trial. We also looked at the motion sensor data, including the only time when the box was actually in motion (i.e., excluding any pauses, see SI). Here we report the results of the behavioral coding since the relationship between uncertainty and exploration may be best indexed by including the time the children could have been planning subsequent actions and thinking about the data they generated, but the primary results hold for all measures (see SI).

To normalize for individual differences in children's exploratory behavior, we computed the time each child spent exploring on each trial as a proportion of the child's total playtime across all four trials and multiplied this proportion by the number of trials in the experiment. Thus, a proportion less than 1 represents less playtime (and a proportion more than 1, more playtime) than would be expected if children distributed their playtime evenly across trials. Although we use proportional playtime to control for individual differences in length of play, all results hold using untransformed (log) playtime reported in seconds (see SI).

To quantify the discriminability of different contrasts, we adopted a variant of the standard signal detection model in which shaking a box with $m$ marbles in it would produce a perceptual trace drawn from some probability distribution over a high-dimensional acoustic space, which can be projected down to a one-dimensional space of abstract numerosity analogous to representations in the approximate number system[44,45]. We modeled the internal representation for each auditorily perceived number as a normal distribution on a log scale (see SI), with equal variances $\sigma$ but logarithmically spaced means, and computed the discriminability of each contrast between $l$ and $m$ marbles presented in Experiments 4–7 in terms of the standard index

$$d' = \frac{|\mu_l - \mu_m|}{\sigma}, \tag{1}$$

where $\mu_l = \log l$ and $\mu_m = \log m$. See SI for a summary of these $d'$ values (Supplementary Table 1), as well as a discussion of alternative ways of estimating discriminability (including different mathematical models, and an empirical estimate from independent adult psychophysical data). These produce nearly identical results for our purposes. We modeled children's intuitions about task difficulty as proportional to this $d'$ measure. Note however those children hear only a single set of marbles in the box on each trial and have no way of judging directly from the auditory data the discriminability of the two set sizes being contrasted. Rather, we posit that children's sense of discriminability depends on their ability to evaluate the contrast between the sounds they hear and their simulation of the sounds they would have heard had the alternative set of marbles been in the box.

Each of Experiments 4–7 was analyzed separately for qualitative effects of discriminability, trial order, and the number of marbles in the box on exploration time (see SI). Here we focus on the preregistered joint analysis addressing our primary question about the effect of discriminability on exploration across all 16 contrasts in Experiments 4–7: Did children systematically explore longer when contrasts were less discriminable? The discriminability of the contrast quantitatively predicted children's exploration time across the full range of contrasts ($\beta=0.24$, 95% CI [0.18–0.30]). Children's exploration time tracked the difficulty of distinguishing the heard and unheard alternative in a remarkably fine-grained way (Fig. 3A, B), correlating strongly with the model whether exploration was coded from video ($r=0.95$; 95% CI [0.78, 0.95]) or with the motion sensor (see SI).

Strikingly, children's exploration time was independent of the number of marbles actually in the box (Fig. 4; $\beta=0.0065$, 95% CI [−0.0094, 0.022]). Thus, although the sensorimotor experience of shaking a box containing only one or two marbles was quite different from shaking a box containing eight or nine marbles, children's exploration depended not only on what they heard but also on what they did not hear: the contrast between the observed evidence and the unheard alternative.

We also analyzed other factors that might affect exploration. Across experiments, children's exploration decreased only slightly over the four successive trials ($\beta=-0.051$, 95% CI [−0.086, −0.016]); age had no effect on children's tendency to explore the hardest contrast longer than the easiest one ($\beta=-0.041$, 95% CI [−0.45, 0.40]). As expected, children's accuracy increased with the discriminability of the contrast ($\beta=1.12$, 95% CI [0.64, 1.46]); there was a marginal effect of age on children's accuracy ($\beta=0.033$, 95% CI [−0.0074, 0.069]).

Finally, we asked whether aggregate behavior in each individual experiment and each individual child's behavior also tended to conform with the predictions of the discriminability model. There was substantial variability in individual children's playtimes, but average playtimes within each experiment were qualitatively well-predicted by a linear fit to the discriminability model (Fig. 5). In addition, in each experiment, a significant majority of individual children explored more, on average, for more difficult discriminations (Fig. 5): for 19/24 children in Experiment 4 (79%; 95% CI [0.58–0.93]), 21/24 children in Experiment 5 (85%; 95% CI [0.68–0.97]), 18/24 children in Experiment 6 (75%; 95% CI [0.53 −0.90]), and 19/24 children in Experiment 7 (79%; 95% CI [0.58–0.9]), a linear regression of that child's playtimes onto

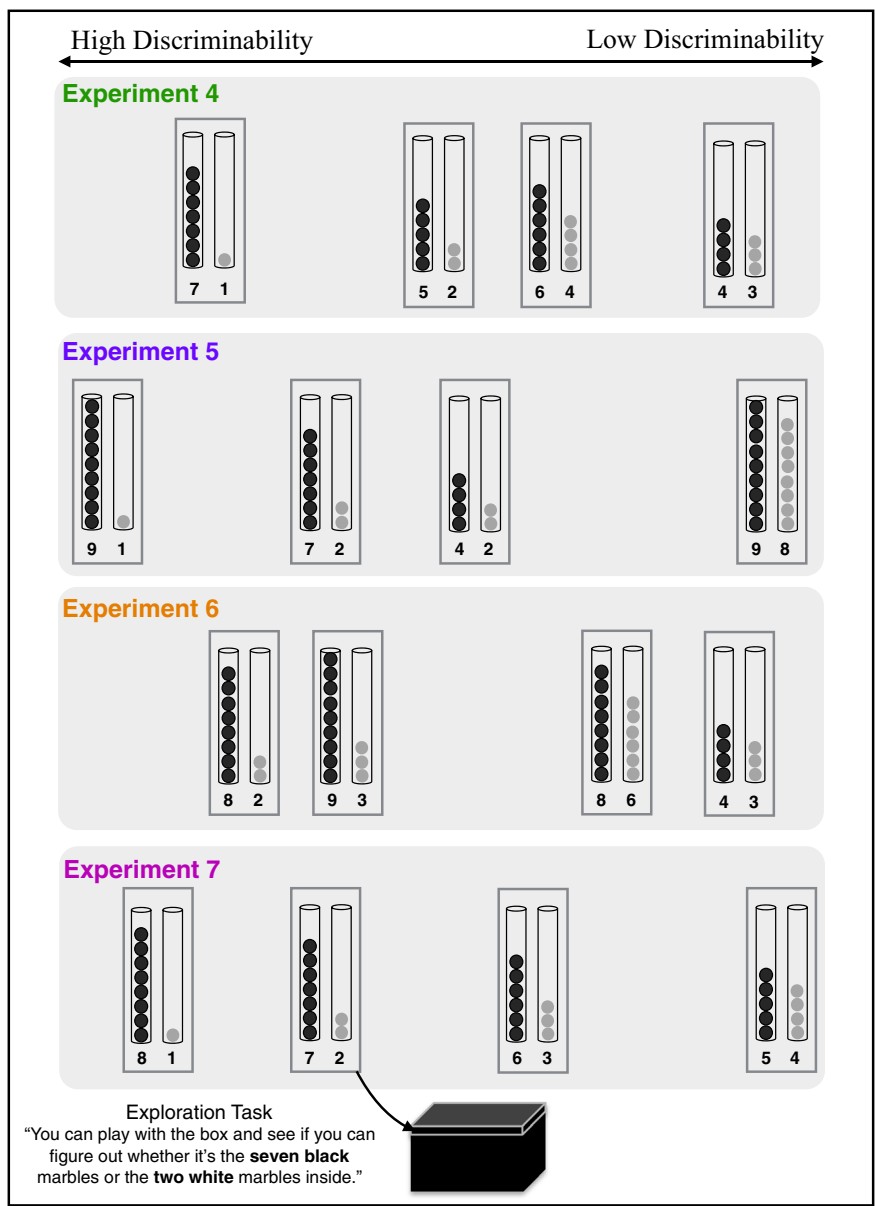

**Fig. 2 Schematic of Experiments 4–7.** The placement of contrasts corresponds to relative discriminability. The actual trial order was counterbalanced, as was the order in which the tubes of marbles were introduced and the contents hidden in the box (e.g., whether 1 or 7 marbles were hidden on the 7-vs.-1 trial), except in Experiment 6, where the content was held fixed at 8 and 3 for both high- and low-discriminability contrasts to provide a within-experiment test of whether content or contrast affected children's exploration time.

discriminability had a positive slope. Hence, not only on average, but at the level of individuals as well, children systematically explored longer when contrasts were less discriminable.

## Discussion

Collectively, the results of these seven experiments suggest that, at least in familiar domains with simple tasks, children can simulate physical interactions and the perceptual data that will result. Furthermore, children can represent their own ability to make the perceptual discriminations needed to compare observed data with simulated, unobserved data under alternative hypotheses. Children represent the relative difficulty of different discrimination problems in ways that support effective decision-making and exploration: they prefer easier problems and explore more given harder ones. The precise, quantitative relationship between

children's exploratory play and the difficulty of perceptual discrimination problems suggests that starting in early childhood, human learners intuitively compute the value of evidence for discriminating alternative hypotheses, and use this sense of uncertainty to rationally calibrate their exploration.

Our account relies on mental simulation, and our quantitative results in Experiments 4–7 analyzed children's exploratory behavior using idealized models of perceptual discriminability in these mental simulations. However, it is possible that children might have relied on some simpler cognitive mechanism or heuristic[46], or a resource-constrained approximation to this ideal[47,48]. One natural alternative to consider for Experiments 4–7 is that children took into account only a simple contrast in the linguistically and graphically presented number of marbles in each pair, without attending at all to the rich perceptual data they obtained in shaking the box or imagining possible sounds they

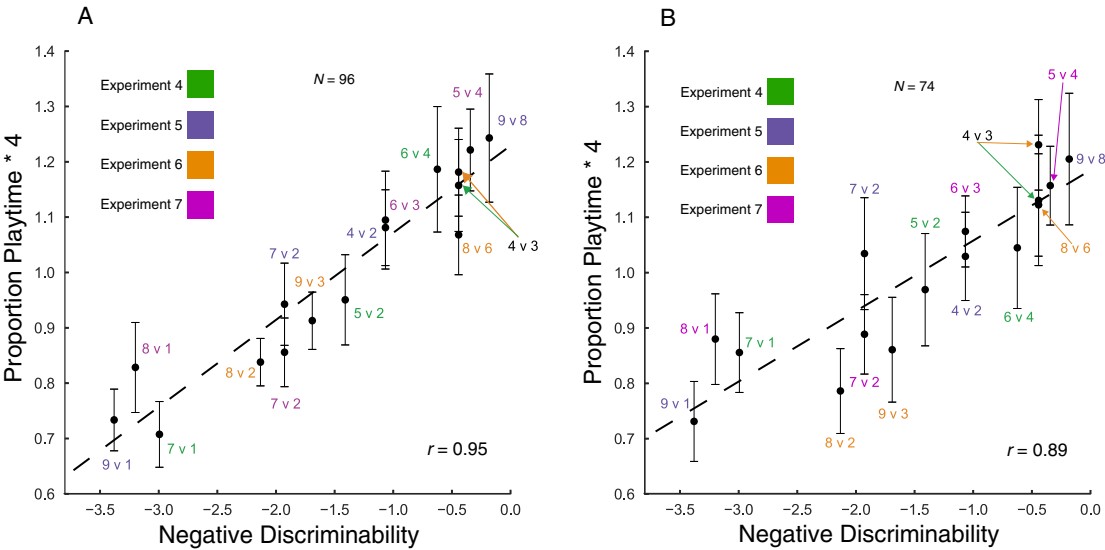

**Fig. 3 Children's proportional exploration times as a function of the negative discriminability of each contrast across Experiments 4–7.** Whether coded by hand (**A**) or by the motion sensor (**B**) children's exploration correlated strongly with the difficulty of the discrimination. Error bars indicate SEMs. Source data are provided as a Source data file.

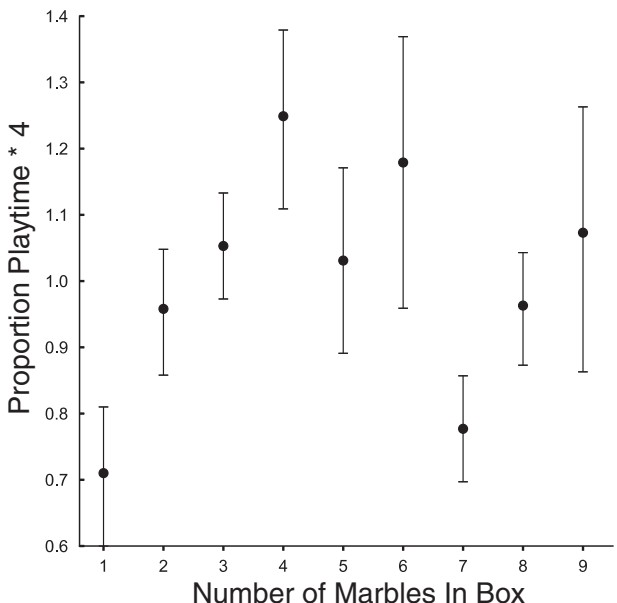

**Fig. 4 Children's proportional exploration times across Experiments 4–7 as a function of the actual number of marbles in the box.** There is no significant correlation between exploration time and number of marbles. Error bars indicate SEMs. Source data are provided as a Source data file.

might hear via mental simulations of box shaking. We evaluated two such heuristic models that avoid the computational burden that might accompany mental simulation, based on the absolute difference and the (negative) ratio of the numbers of marbles in each pair. Both of these models perform well numerically (see SI, Additional Heuristic Models), and so it is indeed possible that children rely on such a mechanism in Experiments 4–7.

The current studies also open up provocative questions for future research. They suggest that children have some metacognitive knowledge about their own ability to make perceptual discriminations. Anecdotally, some children also proffered explicit accounts of their own reasoning. In piloting Experiment 1 for instance, a child said that he preferred the more discriminable

box because the pair was "more not the same". Likewise, in Experiments 4–7, children sometimes explained their own reasoning (e.g., "this one's gonna be hard"). Given the sophistication of the judgment required here (in which children had to compare observed data with unobserved alternatives), we believe children's choices and exploration were less likely to underestimate their reasoning than asking children to justify their choices. However, further research might look at the extent to which children can explicitly account for the reasoning behind their decisions.

Although it seems implausible that children store and retrieve precise representations of the sound of marbles shaken in boxes, we do not know how children (or adults) simulate physical interactions and the sounds they might make with sufficient richness to make these fine-grained discriminations. Intuitively, our ability to imagine what we might perceive given different novel interventions is arbitrarily generative: we can imagine not only how marbles might sound when shaken in a box, but how the sound might change if we added water to the box—or pennies—or a sock. Future work should target both the mechanisms that support these rich online simulations and the limits of our ability to imagine such interactions and their perceivable consequences.

We focused on learners' ability to represent the difficulty of statistical discriminations in a psychophysical context, but our results might reflect a quite general ability to estimate how much data it would take to distinguish competing hypotheses. Future research might look at children's sensitivity to their own ability to discriminate evidence in other domains, probing the extent to which children can engage in these behaviors more broadly.

We also do not know to what extent the abilities children showed here might emerge earlier in development, or in non-human animals. When confronted with easy and difficult problems, children as young as three adapt their behavior appropriately (i.e., opting out of difficult problems or asking for help[38]; future research might look at whether young preschoolers —or in simpler contexts, even toddlers and infants—might, as here, also be able to anticipate the relative difficulty of different kinds of problems and adjust their choices and exploration accordingly. Similarly, macaques, capuchins, apes, and dolphins show some sensitivity to their uncertainty across a range of tasks (see ref. [49] and refs. [50,51] for reviews and discussion); the current paradigm might be adapted to test intuitive psychophysics across

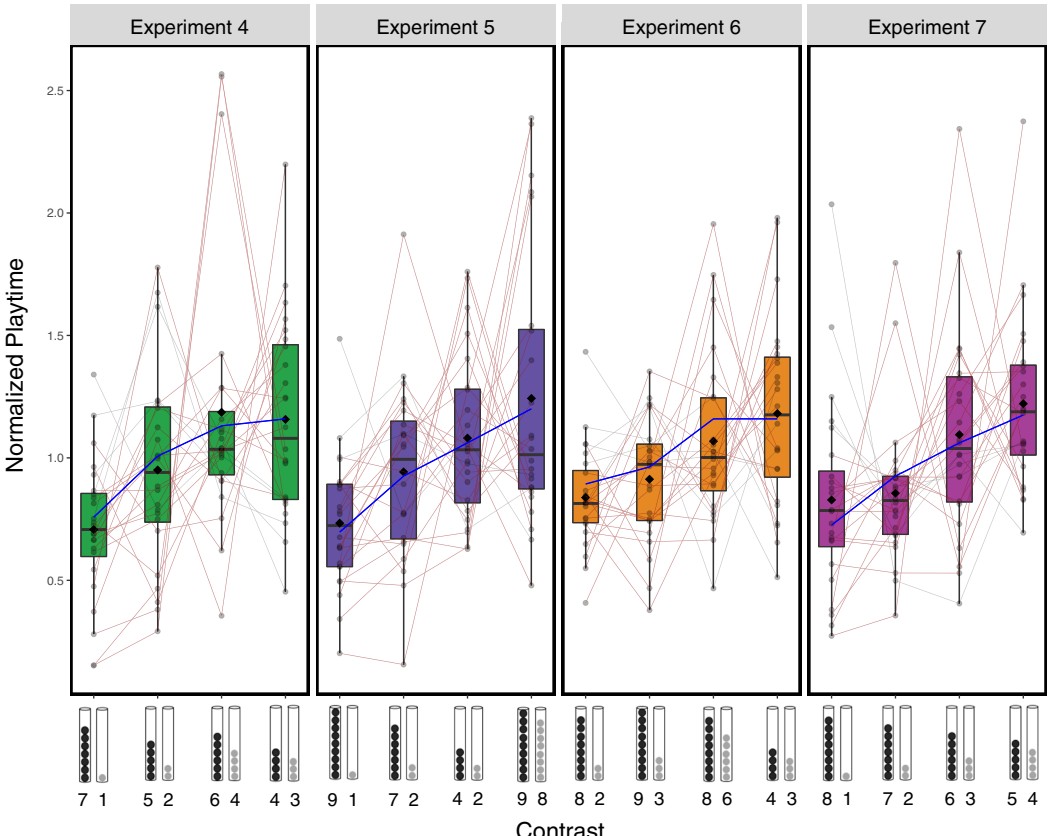

**Fig. 5 Behavior of individual children (normalized playtimes) on each condition of Experiments 4–7.** Conditions ordered by discriminability ($n = 24$ per experiment). Diamonds represent condition means, and box plots indicate medians, 25th and 75th percentiles, and outlier ranges. Blue lines show the predictions of the discriminability model under a linear fit to mean playtimes. Thin lines connect the responses of each individual child, with red lines indicating children who qualitatively followed the model's predictions, exploring more on average when contrasts were harder (i.e., a linear regression of that child's playtimes onto discriminability had a positive slope). Source data are provided as a Source data file.

species. Would, for instance, a nonhuman primate be able to infer the probable contents of a container from the sound it made when it was shaken? If two containers were shaken and the animal heard a sloshing sound, would it preferentially open the box which could have contained the juice or a rock, or rather than the one which could have contained juice or water? Queries like these might allow us to test the extent to which our ability to recover the generative causes of perceptual stimuli, compare heard and unheard alternatives, and prefer more discriminable evidence emerges across species.

Finally, here we probed children's ability to reason back a single step in a causal chain: from the sound objects made when shaken in a box to the objects making the sound. But as lay adults, we can reason backward through multiple steps in a causal chain to events increasingly remote from direct experience. We can see the lights go out and infer that a storm knocked over a tree branch and downed a power line, or we can see a pileup of traffic and infer that a ship is passing under a drawbridge, miles up the road. Our work suggests that young children can go from perceptual data to the physical causes that gave rise to them, and compare their observations with other evidence they might have observed, in order to make rational choices about how to explore. Future work might look at how these intuitive capacities develop into ones that can guide learning and discovery over a lifetime, culminating in the scientific practices that let us connect observations to events that are too big or too small, too fast or too slow, or too remote in space or time for direct perception. Progress on these questions has the potential to give us new insight into the origins of inquiry.

## Methods

**Participants**. Across Experiments 1–7, we recruited 184 children (mean: 5;2, range 3;0–8;6) who were visiting a local children's museum. Sixteen other participants were excluded from the analysis due to preferring the distractor object[12], experimenter error[3], failure to pass inclusion trial or attend to task[4], and family interference[1]. All experiments were approved by an institutional review board for human subjects and all ethical guidelines were followed. The child's parent or legal guardian was provided with a verbal description of the study. The experimenter answered any questions the parent had. The parent or legal guardian then provided written informed consent to participation and videotaping of the study consistent with the MIT IRB approval for the study. Children over age seven also provided verbal assent to participate.

**Materials**. In all preliminary studies, two cardboard shoeboxes covered with black electrical tape were used and a large cardboard screen (80 cm × 60 cm) was used as an occluder. In the *Object Identity* study, a square beanbag and a plastic ball of equal weight were used (5-cm diameter). For all other preliminary studies, ten colored marbles and two translucent cylindrical tubes were used. A stuffed animal bunny was used as a character in the script. In the *Volume Control* experiment, a felt cloth fitted to the bottom of the shoebox was used to alter the sound of the marbles when shaken.

For Experiments 1–3, the same tape-covered cardboard boxes and screen were used as in the preliminary studies, with the items being hidden differing between experiments. In Experiment 1, two pencils with a shiny, holographic coating were used as target objects. A standard yellow pencil and a small, cotton-filled fabric cushion were used as distractor objects. In Experiment 2, one large (approximately 8 cm by 5 cm) and six small (approximately 3 cm by 2 cm) plastic elephants were used. A small plastic pig (approximately 3 cm by 2 cm) was also used. A transparent, hexagonally partitioned container was used as the baby elephants' home. In Experiment 3, four transparent cylinder tubes were used. Two tubes each contained eight different colored marbles, arranged to look identical to each other; one tube contained two white marbles, and one tube contained six white marbles. The tubes were sealed at the top with packing tape. Drawings of each of the marble tubes were also used as a memory cue. A stuffed animal bunny was used to occupy

the children's hands so that they did not reach for the stimuli or interfere with the demonstrations.

In Experiments 4–7, a single tape-covered shoebox (18 cm × 16 cm x 12 cm) was used. Four objects were used in the practice trials: a plastic duck, a star-shaped pillow, a flat glass bead, and a cotton ball. For the test trials, standard-size glass marbles in eight colors and eight translucent cylindrical tubes were used. The tubes were preloaded with the appropriate number of marbles and sealed at the top; although children were told that the tubes of marbles would be poured into the box, marbles were in fact added quietly by hand to ensure that children did not get any evidence about the sound until they themselves shook the box. A large cardboard screen (80 cm × 60 cm) was used both as an occluder and as an answer board with six Velcro tabs for children to provide their responses. Laminated pictures with Velcro tabs on the back, approximately to scale, were used to depict the possible contents of the box for both the practice trials and the test trials.

All children were tested individually in a private testing room off of the museum floor. The child and the experimenter sat on opposite sides of a child-sized table. All sessions were videotaped. Children's responses were coded live by the experimenter and recoded by a coder blind to conditions from video. In addition to measuring children's exploratory behavior via video coding, we developed an independent measure based on the time course of the motion of the box. We equipped a microcontroller with an accelerometer and placed the device in a small compartment of the box (the compartment was attached at the top corner of the box so as to minimize the possibility that it might interfere with box shaking). Custom software wirelessly transmitted the accelerometer readings, in real time, to a computer that recorded the measurements. The experimenter pressed a button at the start and end of every trial to record the time interval during which box shaking could have occurred.

**Reporting summary**. Further information on research design is available in the Nature Research Reporting Summary linked to this article.

## Data availability
A reporting summary is available as a Supplementary information file. All data are available at https://osf.io/n97fr/. Source data underlying Figs. 3–5 are available as a Source data file. Source data are provided with this paper.

## Code availability
All source code for analysis is available at https://osf.io/n97fr/.
    *See SI for detailed materials, methods, and procedures.*

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

## Acknowledgements

We thank the Boston Children's Museum and the families who participated in this research. We also thank Nancy Kanwisher, Josh McDermott, Drazen Prelec, and Rebecca Saxe for reviewing drafts of the paper, Angela Kim and Julia Simon for help with data collection, Kary Richardson for coding, Kevin Smith and Julian Jara-Ettinger for statistical assistance, and Regina Ebo for assistance with the references. This material is based on work supported by the Center for Brains, Minds, and Machines, funded by NSF STC award CCF-1231216 and an NSF Graduate Research Fellowship to R.W.M.

## Author contributions

R.W.M. assisted with the study design, piloted Experiments 1–3, ran Experiments 4–7, and contributed to the data analysis and writing; M.H.S. conceived of the study, ran the preliminary experiments and Experiment 1, developed the model, and contributed to the data analysis and writing; M.P. ran Experiments 2–3 and contributed to the data analysis and writing; J.B.T. contributed to the study design, model, and writing; L.E.S. contributed to the study design and writing.

## Competing interests

The authors declare no competing interests.
