## [Peer Review File · Nature Communications]

Reviewers' comments:

Reviewer #1 (Remarks to the Author):

This paper is well written, and uses clever experimental designs to examine the ability to discriminate "hypotheses". Please see below for questions and comments.

These are intriguing results, but I am not convinced that they are especially novel. The design and the particular outcome measures are new (esp re Expts 4-7), but the finding that children of this age are exceptionally good at solving problems of this sort is very well established. Children are even good at providing explicit reasoning about their judgements (explanations) for these sorts of problems at this age...

On that note, why not ask participants to explain their behavior?

How is intuitive psychophysics different from intuitive physics, which we know infants have?

Why not include 2 and 3 year olds?

Would you predict the same results from older children and adults in Expts 4-7?

I have a number of questions about sample size, which is quite small in each of the experiments conducted. How did you decide how many participants to include in each study?

Reviewer #2 (Remarks to the Author):

Siegel et al present an interesting series of experiments that investigate children's intuitions about the kinds of observed data that would be produced by alternate hypotheses (Experiments 1-3) and that investigate children's behavior in perceptual discrimination tasks of varying difficulty (Experiments 4-7). Results suggest that children have the ability to reason about alternative hypotheses, perhaps to update their beliefs about the hypotheses' plausibility given perceptual evidence obtained, and to make a decision accordingly.

Siegel et al contextualize the work with themes of intuitive psychophysics, information search, and the intuitive scientist metaphor.

I think the experiments and results are interesting and, with appropriate revision, the manuscript could be both interesting and accessible to a wide audience. The Introduction and General Discussion sections nicely explain why the work is interesting and important. The manuscript is well written.

The fact that my review is long does not diminish that. All good work has the ability to be improved.

My main top-level concern, which can be addressed in a conscientious revision, is to be more meticulous in differentiating between what the authors actually tested, and the reasons they (and we all) should think that the work is interesting. One example: The intuitive psychophysics idea is interesting. I think it may be a bridge too far at this point to claim that the results obtained substantiate this conclusion. I think there are a number of small tweaks to the text that would make clear that this idea is one reason the work is interesting, without claiming that the data to date substantiate this conclusion. For instance, the title could be tweaked to be "Intuitive psychophysics? Children's exploratory play tracks the discriminability of hypotheses". Changing a colon to a question mark is an example of a very small tweak that would preserve the hook to grab diverse readers' attention, while perhaps being more faithful to the level of evidence thus far obtained. I would also take the statement in lines 66-67 and reword it as a question, which could also be more motivating to the reader than seeing such a statement already in the introduction section. Another example: Experiments 1-3 are described as "Deciding what to explore". I agree (for me at least) that deciding what to explore is an important problem. The authors could make a careful case for how these experiments test abilities that the authors believe are important constituent abilities for choosing informative things to test or explore. The immediate tasks are not (to my mind) really exploration; they are (very interesting) decision problems that require some ability to reason (a la Raiffa's pre-posterior analyses) about exploration. An exploration variant (which I would also find interesting) could involve having the children select which of two boxes to shake, or to simply let them shake the boxes (and measure which box is shaken first, how long or for what proportion of the time, etc). I don't think the authors need to do that experiment now, but they should be more careful in discussing what they did, as opposed to why they think it is interesting.

The manuscript would already be quite great as a developmental paper, but for this journal I think it is important to connect to other strands of relevant literature (in science more broadly) and in the history of psychology. This is all very doable, and will broaden the range of readers who notice and appreciate the work here. More specifically:

Experiments 4-7 are very similar to the optimal stopping (or determining the size of the sample) experiments conducted by Wald and colleagues. For reviews see the 1967 Peterson and Beach Psych Bulletin paper on Man as an Intuitive Statistician. Craig McKenzie posted it at <http://pages.ucsd.edu/~mckenzie/Peterson&Beach1967PsychBulletin.pdf>; see page 37 on Determining the Size of the Sample. The earlier work was not (that I can remember) explicitly in a perceptual domain, but sets a nice foundation for the current work.

The intro nicely grounds the investigation in scientific inference, in particular the idea of discriminating among multiple hypotheses. It would be nice to cite at least a couple of classics in this area, like Chamberlin's (1897) Method of Multiple Working Hypotheses (<https://science.sciencemag.org/content/148/3671/754>), or Platt's (1964) Strong Inference (<https://science.sciencemag.org/content/146/3642/347>). There is also a lot of work in statistics and

philosophy of science (Good, 1950; Lindley, 1956; Fedorov 1970s) that attempts to instantiate these ideas in a probabilistic Bayesian context, which would be relevant when thinking about the present Experiments 1-3. Beyond the developmental literature (which is nicely cited) there is other recent work in psychology (the most famous being Oaksford and Chater's 1994 Selection Task model, <https://psycnet.apa.org/record/1995-08271-001>; Coenen et al.'s 2018 paper is a current overview/review, <https://doi.org/10.3758/s13423-018-1470-5>).

The article, at several points, dismisses the idea that children could be using simple heuristics to achieve their performance on the tasks. There is a great deal of work on people's use of simple heuristics (some of it even focusing on children), and recently on trying to build models that would make purely Bayesian (Marr's computational level) more psychologically plausible (bounded rational / Falk Lieder style). Why is none of this work (neither Gigerenzer-esque nor Griffiths/Lieder-esque) cited or considered in a serious way? It does not need to be in opposition to the more computational-style rationale for doing things in a particular way. As an example, the authors provide a very complex rationale for their model of perceptual discriminability from which d' is derived, as a basis for analyzing the results from Experiment 4-7. I trust this (and Josh McDermott is in the acknowledgements). I really wanted to know what the kids were doing. The authors must have some insight on this, after so many children did the task, videotaped, tested on an individual basis. I'm thinking of things like checking how far away from a 50:50 split the marbles are, and allocating observation time (shaking and listening) according to that. I would hope that a little bit of the authors' insight on this (it can be qualitative), after testing so many children, could be included in the article proper. Certainly it would be sensible to include a section in the Supplemental material on the authors' best, if tentative, ideas about the kinds of strategies that children used on the various tasks.

The biggest disappointment in terms of the experiments conducted, relative to the expectations that the title and abstract of the article set up, is that we have no direct evidence for children's ability to track their own perceptual abilities. I think the paper is in any case really interesting, but could be framed in a way that readers are not similarly disappointed. Alternately, a straightforward extension of Experiments 4-7 would be to "ask" children how "hard" a particular discrimination task is. Is it easier to tell whether a the Blue box has 1 or 9 marbles, or whether the Green box has 2 or 8 marbles?. What if the Blue box has 3 or 5 marbles, and the Green box has 6 or 8 marbles? Questions along these lines (adapted to a method that will work with children of the desired age) could provide more direct evidence for the idea (as claimed in the abstract) that children "represent their own perceptual abilities". Of great interest to test would be scenarios in which different possible heuristic strategies (my proportion-based heuristic, a heuristic that just looks at the difference in marble counts, etc.) contradict each other and/or the discriminability model.

One small point is that the manuscript at several points notes that the data analysis was conducted by individuals who were blind to the experimental conditions. This is great. But isn't the coding of this (which box a child selected, how long a child shook a box) relatively objective? The bigger concern for me would be that due to the design of the experiment, the experimenter could not be blind to the experimental conditions. I don't think that should hold up the paper, but I think it would be helpful and

appropriate to explicitly acknowledge this in the article, and perhaps to comment further in the Supplemental material on the rationale (or necessity) of the experimenter not being blind, why it should not matter, etc.

Other suggestions

Figure 1 and the description of the experiments in the body of the text were not adequate for me. The key point missing, which needs to be explicit in the figure, is that "each box contained exactly one item". I think this figure could be redone in a somewhat more helpful way. It can be done visually, but I illustrate how it might work schematically below:

Experiment 1. Box 1 contains a shiny pencil or a pillow. Box 2 contains a shiny pencil or a boring pencil.

Experiment 2. Box 1 contains a small elephant or a large elephant. Box 2 contains a small elephant or a small pig.

(etc.)

I think it would also be helpful to outline the basic experimental procedure in this figure or in a caption to this figure. Something like "Box 1 contains either the shiny pencil or a pillow. Box 2 contains either a shiny pencil or a boring pencil. Both boxes, when shaken, made sounds consistent with there being a pencil inside. You get to pick one box, open it, and keep the item that is inside it. Which box would you like to open?".

If children were told a little white lie, don't use euphemisms "children believed X but actually Y was true" to describe it. Just state what children were told and what was in fact the case.

Make sure that the paper is meaningful to people who are not developmental or cognitive psychologists. There are things like the years;months of age convention, which are not generally known, but are used without explanation in the manuscript.

Test the method description (in the body of the manuscript) plus figure captions with someone who is in a different area (e.g. a biophysicist who does not do anything cognitive) to see if they can understand it. I found it hard, from the manuscript, to figure out what exactly was done. I think I did in the end, but it required a couple of iterations through the manuscript to do so; that should not be the case.

The abstract needs proofread and rethought. It should give a better idea of what was done and why it is important.

I would like to see an acknowledgement of the fact that the experimenters were presumably not blind to the condition, and why the authors believe that that is not a problem. (Hans the horse effects).

Suggestions with respect to specific parts of the manuscript follow.

Abstract

If the authors (as I hope) are serious about the idea that testing among hypotheses should be the goal in science, it is funny to state that the value of evidence is "for testing a [single] hypothesis". The whole idea, starting at least with Chamberlin's (1890s) work, more recently with Good, Lindley, Platt, and others, is that scientists should figure out (and have as their goal) how to discriminate among many hypotheses, rather than simply to test a single hypothesis. I would delete "for testing a hypothesis" in the first sentence.

Please proofread the first sentence of the abstract. Either insert "that" before "are under" or delete "are".

"Children's exploration time was independent of the evidence heard". This is hard to understand, in the abstract. I think the sentence can be written in a clearer way, perhaps simply "Children's exploration time, across 16 contrasts, quantitatively tracked the discriminability of heard evidence from an unheard alternative".

Intro

[line 54] "subjective discriminability of competing hypotheses". The word "subjective" here put my off on the first read, and I actually think it is not consistent with the experiments conducted, or at best is confusing. My understanding of the d' and other discriminability models used in Experiments 4-7 is that they are intended to be objective models of how hard particular discriminations are for people to make. "Subjective", to me, refers to people's (in this case children's) understanding of how difficult a particular discrimination will be.

[lines 75-75] I'm not sure what "looked only at qualitative relationships between children's uncertainty and exploration" means. Certainly these studies quantified the things they were measuring, and manipulated uncertainty in various ways.

Figure 1. It would help me if a pseudocode-style version of the experimental procedure were included here. In particular, it would be helpful to make clear that each box contained exactly one item. (Not everyone is working in this research area, and it does not go without saying that a box contains exactly one target.)

[lines 200-201]. The proportional playtime statistic seems to be a reasonable dependent variable to measure. It is helpful to note that results also hold if using log playtime. Did the authors also test using raw playtime?

Figure 5. Please check the within-figure headings: I think Experiment 5 through 8 should be numbered Experiment 4 through 7 to match the numbering elsewhere in the paper.

Methods

[lines 373=375] "although children believed the tubes of marbles". Weren't the children told this? It is better to state what is known, namely what instructions the children were given. We don't (I think) know what the children believed. If there were some "white lies" along the way to make the experimental procedure work, it is really better to just state that clearly, because this kind of wording is hard to parse. Please check the manuscript for this; I think similar things came up a couple of other places (e.g. in the Supplementary Materials section).

References

A nice set of reference. It would be helpful to explicitly connect to some older and other strands of relevant work.

Supplemental Materials

It was hard to get to know the experimental methodology. Referring explicitly back to the figures in the main text, where appropriate, when describing experiments in the Supplemental materials, would be helpful. Figures 1 and 2, as previously noted, could be expanded to provide some more helpful information. If space constraints preclude doing this in the main text (which I doubt-- I think you can do it), then comprehensive figures to describe the procedure should be included in the Supplemental materials.

[line 578] do you mean to italicize Volume Control, to be consistent with naming of other preliminary experiments?

[lines 832-840] It is important to state, in numbers, what happened with the "pre-registered additional sample of 24 children". It can leave the reader with a bad impression if you decline to provide explicit numbers in cases where the numbers are unhelpful. Just state what happened.

I sign my reviews. I should also acknowledge my PhD student Lara Bertram who (with the editor's permission) meaningfully contributed to this review.

Jonathan Nelson

Reviewer #1 (Remarks to the Author):

This paper is well written, and uses clever experimental designs to examine the ability to discriminate "hypotheses". Please see below for questions and comments. These are intriguing results, but I am not convinced that they are especially novel. The design and the particular outcome measures are new (esp re Expts 4-7), but the finding that children of this age are exceptionally good at solving problems of this sort is very well established.

We appreciate your interest in the studies. We agree that many aspects of the sophistication of children's causal reasoning are indeed well-established: children distinguish spuriously associated events and causal relationships, integrate evidence with their intuitive theories, and selectively explore when evidence is surprising or ambiguous (see e.g., Gopnik & Wellman, 2012; Schulz, 2012 for reviews). However, we respectfully disagree that any previous work has investigated, or shown, the kind of reasoning we show here. In contrast to previous studies, here the evidence needed to distinguish competing hypotheses is both available and, in principle, informative: what is at stake is children's ability to understand their own threshold for discrimination. That is, here we look at children's understanding not of whether evidence is or is not informative, but how readily they themselves can process the available evidence.

Furthermore, in contrast to studies that measure children's actual ability to make perceptual discriminations (e.g., in distinguishing small or large approximate number ratios; Vo, Li, Kornell, Pouget, & Cantlon, 2014), here we ask children to compare observed data with a simulation of data they do *not* observe. If children recognize that they should preferentially open a box that might have contained eight or two marbles rather than one that might have contained eight or six marbles (as in Experiment 3 of our study, and the logic is similar in Experiments 1 and 2) they can only do so by comparing the observed evidence for eight marbles with the difficulty of the counterfactual discriminations. No other study has suggested that children can engage in this kind of comparison with simulated data. Finally, no previous work has shown (as we do in Experiments 4-7) that children's exploration quantitatively tracks the discriminability of the hypotheses. We apologize for failing to clarify the novelty of the current contribution in the previous version of the manuscript. We now emphasize these distinctions as follows (lines 80-93):

In particular, previous studies looking at children's sensitivity to their own uncertainty have considered cases where evidence is surprising (e.g., 47-48), uninformative with respect to competing hypotheses (e.g., 49), or cases where children simply do not know answer to a query (e.g., 50-52). In contrast, here we look at cases where evidence to distinguish hypotheses is available and, in principle, informative, and we ask whether children represent their own ability to make distinctions among the available evidence. Specifically, rather than asking whether children can distinguish two different observations (as one might in a psychophysics experiment), we allow children to observe only one kind of event and we ask whether they recognize that that observation is more discriminable from some hypotheses than others. That is, we are interested in whether children can simulate the evidence they might get under alternative hypotheses and compare the discriminability of observed evidence with unobserved alternatives. Finally, we ask whether there is a precise

quantitative relationship between the discriminability of competing hypotheses and children's active exploration.

Children are even good at providing explicit reasoning about their judgements (explanations) for these sorts of problems at this age... On that note, why not ask participants to explain their behavior?

There has been work on children's metacognition across a range of tasks showing that even very young children are aware of when they have and have not forgotten information or when they are uncertain about the answer to a query (see e.g., Lyons & Ghetti, 2010 for review), and as the reviewer notes, elementary school-age children can sometimes articulate their reasons for believing one claim over another (e.g., Sandoval & Cam, 2010). However, as noted above, these tasks are in several key respects, less sophisticated than the task we ran above. In particular, our discriminations vary in difficulty without imposing any burden on children's memories and without providing children with either surprising or uninformative data. To the degree that children can explicitly reason about why they nonetheless think some of these judgments will be harder than others they have to be able to explicitly note the contrast between the evidence they heard and what they might have heard instead. Given that no prior work had demonstrated that children could make these kinds of judgments even implicitly, we did not ask whether children could explicitly justify their reasoning.

However, even though we did not explicitly ask for verbal reports of children's reasoning, children sometimes spontaneously commented on the tasks in ways that seem revealing. For example, in piloting for Experiment 1, a child commented that he preferred the more discriminable box because its object pair was "more not the same". Similarly, in Experiments 4-7, children sometimes made comments that seemed informative – saying "this one's gonna be hard" when confronted with a difficult discrimination, for example. We did not wish to rely on an explicit explanation because children could well have the ability to succeed in these tasks without having the linguistic ability to account for their own judgments; children's choice behavior and exploration was less likely to underestimate their competence in these tasks. Nonetheless, we agree with the reviewer that children's explicit insight into their own reasoning is of great interest, and even anecdotal evidence is worth noting. We now discuss this in the manuscript (lines 346-356) as follows:

The current studies also open up provocative questions for future research. They suggest that children have some metacognitive knowledge about their own ability to make perceptual discriminations. Anecdotally, some children also proffered explicit accounts of their own reasoning. In piloting Experiment 1 for instance, a child said that he preferred the more discriminable box because the pair was "more not the same". Likewise, in Experiments 4-7, children sometimes explained their own reasoning (e.g., "this one's gonna be hard"). Given the sophistication of the judgment required here (in which children had to compare observed data with unobserved alternatives), we believe children's choices and exploration were less likely to underestimate their reasoning than asking children to justify their choices. However, further research might look at the extent to which children can explicitly account for the reasoning behind their decisions.

How is intuitive psychophysics different from intuitive physics, which we know infants have?

Insofar as children have an intuitive physics, they can make judgments about the physical world (e.g., recognizing that unsupported objects will fall, that objects won't pass through solid walls, that moving objects can displace other objects on contact, etc.). Abundant research suggests that these abilities are present in infancy. Similarly, infants can make judgments about what other agents see, hear, want and know— these abilities are referred to as an intuitive psychology. Quite separately, scientists have looked at children's psychophysical judgments: their ability to discriminate perceptual stimuli (as in hearing tests or vision tests).

The ability we refer to here as an “intuitive psychophysics” is distinct from all of the above. We ask whether children represent their *own* ability to discriminate perceptual stimuli. In our tasks, we presume upon children's intuitive physics (i.e., that children can represent the physical interactions among objects in a box) and their intuitive psychology (i.e., assuming that children represent the relationship between seeing and knowing). But children could easily have an intuitive physics, an intuitive psychology, and make psychophysical discriminations themselves (hearing the difference between two quite different sounds and failing to hear the difference between two similar ones) without *knowing* that they will have an easier time distinguishing sounds to the degree that they are different. Our study demonstrates, for the first time, that children represent their own ability to make perceptual discriminations.

We apologize for being unclear about this distinction in the previous version of the manuscript. We now discuss the distinction among intuitive physics, intuitive psychology, psychophysics and intuitive psychophysics in detail as follows (lines 57-71):

Our proposal builds on three more basic capacities that we already know children possess: aspects of intuitive physics (i.e., the ability to represent the physical interactions among objects) and intuitive psychology (i.e., the ability to represent the relationship between seeing and knowing), and an ability to make psychophysical discriminations themselves (i.e., to hear the difference between two quite different sounds more easily than the difference between two similar ones). In asking whether children have an “intuitive psychophysics”, we are asking whether children can use these abilities to judge whether they themselves will be able to distinguish evidence for different physical interactions. Can children simulate the interactions among physical events and the perceptual consequences of these interactions with sufficient granularity to represent their own ability to discriminate among events? Note that having an intuitive psychophysics need not imply that children can explicitly explain or justify their own judgments (any more than having an intuitive physics requires that children be able to explain their own reasoning about objects and forces). However, to the degree that children have an intuitive psychophysics, they should be able to represent the relative difficulty of discriminating perceptual evidence and these representations should guide their judgment and exploration.

Why not include 2 and 3 year olds?

We included two and three-year-olds in the preliminary experiments where we simply asked whether children could use the acoustic stimuli to identify which of two objects (e.g., a plastic

ball or a soft beanbag) were in the box. Even two-year-olds were able to reason back from the sound of an object to its identity. However, two-year-olds could not meet the task demands Experiments 1-3 (which required them to understand, among other things, that one of two items could be placed in each box). Three-year-olds were included in the range of Experiments 1 and 2. But the remaining experiments involved numerical comparisons between sets of marbles. Since we were more interested in whether young children could represent their own perceptual abilities at all than in establishing the youngest age at which this ability emerged, we excluded three-year-olds from Experiments 3-7 where the ability to represent the numerosity of different sets of marbles accurately was a pre-requisite to the task. We now note this in the Supplemental Information as follows (lines 751-754):

Although we included two-year-olds in the preliminary experiments, we did not include them in the following studies because pilot work established that the task demands (requiring them to represent that one of two items could be placed in each box) were too high.

(and lines 869-871):

We restricted the age range to children four and up in this and the following experiments because accurate numerosity judgments were critical to the tasks and three-year-olds' ability to count is fragile (e.g., 10).

Would you predict the same results from older children and adults in Expts 4-7?

Yes, in particular because our analysis showed no relationship between age and children's tendency to explore the hardest contrast longer than the easiest.

I have a number of questions about sample size, which is quite small in each of the experiments conducted. How did you decide how many participants to include in each study?

The effect sizes in the preliminary studies were relatively large so the sample size was correspondingly small. We report the results of the power analyses throughout in the Supplemental Information (lines 802-809):

Experiment 2

Participants

Based on the results of the preliminary experiments, we estimated the effect size for a single experiment as $f = 0.29$. We used the power calculation program G*Power to calculate the planned sample size of for this experiment using $f = 0.29$, $\alpha = 0.05$, and power = 0.80. The projected sample size using these values is 24 participants, which was used for Experiments 2 and 3.

(and lines 951-958):

Experiments 4-7

Experiment 4

Participants

Participants were recruited from an urban children's museum. Consistent with the previous studies, we estimated the effect size (f) for a single experiment was .29. We used the power calculation program, G*Power, to calculate the planned sample size of for this experiment using $f = 0.29$, $\alpha = 0.05$, and $\text{power} = 0.80$. The projected sample size using these values is 24 participants. Twenty-four children (mean age = 5;9; range 4;1-8;2) were included in the final sample. One additional child was excluded because they did not explore before providing a response on one or more trials (see Procedure for details).

Reviewer #2 (Remarks to the Author):

Siegel et al present an interesting series of experiments that investigate children's intuitions about the kinds of observed data that would be produced by alternate hypotheses (Experiments 1-3) and that investigate children's behavior in perceptual discrimination tasks of varying difficulty (Experiments 4-7). Results suggest that children have the ability to reason about alternative hypotheses, perhaps to update their beliefs about the hypotheses' plausibility given perceptual evidence obtained, and to make a decision accordingly.

Siegel et al contextualize the work with themes of intuitive psychophysics, information search, and the intuitive scientist metaphor.

I think the experiments and results are interesting and, with appropriate revision, the manuscript could be both interesting and accessible to a wide audience. The Introduction and General Discussion sections nicely explain why the work is interesting and important. The manuscript is well written.

The fact that my review is long does not diminish that. All good work has the ability to be improved.

My main top-level concern, which can be addressed in a conscientious revision, is to be more meticulous in differentiating between what the authors actually tested, and the reasons they (and we all) should think that the work is interesting. One example: The intuitive psychophysics idea is interesting. I think it may be a bridge too far at this point to claim that the results obtained substantiate this conclusion. I think there are a number of small tweaks to the text that would make clear that this idea is one reason the work is interesting, without claiming that the data to date substantiate this conclusion. For instance, the title could be tweaked to be "Intuitive psychophysics? Children's exploratory play tracks the discriminability of hypotheses". Changing a colon to a question mark is an example of a very small tweak that would preserve the hook to grab diverse readers' attention, while perhaps being more faithful to the level of evidence thus far obtained. I would also take the statement in lines 66-67 and reword it as a question, which could also be more motivating to the reader than seeing such a statement already in the introduction section.

The Reviewer returned to this point later in the Review so we include that here:

The biggest disappointment in terms of the experiments conducted, relative to the expectations that the title and abstract of the article set up, is that we have no direct evidence for children's ability to track their own perceptual abilities. I think the paper is in any case really interesting, but could be framed in a way that readers are not similarly disappointed. Alternately, a straightforward extension of Experiments 4-7 would be to "ask" children how "hard" a particular discrimination task is. Is it easier to tell whether a the Blue box has 1 or 9 marbles, or whether the Green box has 2 or 8 marbles?. What if the Blue box has 3 or 5 marbles, and the Green box has 6 or 8 marbles? Questions along these lines (adapted to a method that will work with children of the desired age) could provide more direct evidence for the idea (as claimed in the abstract) that children "represent their own perceptual abilities".

We appreciate the reviewers' feedback. We have adopted all of these suggestions. We have now titled the manuscript:

Intuitive psychophysics? Children's exploratory play tracks the discriminability of hypotheses

And we have changed the statements proposing that children have an intuitive psychophysics to questions, as follows (currently lines 52-56):

Here we ask whether such an "intuitive psychophysics" guides children's exploration. Can children use their intuitive understanding of both the physical world and their own ability to make perceptual discriminations to engage in effective exploration? Do they compare the perceptual evidence they observe with the evidence they think they would have observed under different competing hypotheses?

We have also modified the abstract to note that the results are simply consistent with the idea that children have an intuitive psychophysics (lines 21-23):

The results are consistent with the idea that children have an *intuitive psychophysics*: they represent their own perceptual abilities and explore longer when hypotheses are harder to distinguish.

Perhaps most critically however, we now clarify what we think is entailed in an intuitive psychophysics. We agree with the Reviewer that it would be interesting if the children could offer explicit justifications of their judgments. However, as detailed in response to Reviewer 1 above, we do not believe these explicit linguistic justifications are the best test of children's abilities. Children might well represent their own ability to make perceptual discriminations without being able to explain to an experimenter the basis on which they make these judgments. For the sake of completeness, we re-excerpt the relevant passages (lines 57-71):

Our proposal builds on three more basic capacities that we already know children possess: aspects of intuitive physics (i.e., the ability to represent the physical interactions among objects) and intuitive psychology (i.e., the ability to represent the relationship between

seeing and knowing), and an ability to make psychophysical discriminations themselves (i.e., to hear the difference between two quite different sounds more easily than the difference between two similar ones). In asking whether children have an “intuitive psychophysics”, we are asking whether children can use these abilities to judge whether they themselves will be able to distinguish evidence for different physical interactions. Can children simulate the interactions among physical events and the perceptual consequences of these interactions with sufficient granularity to represent their own ability to discriminate among events? Note that having an intuitive psychophysics need not imply that children can explicitly explain or justify their own judgments (any more than having an intuitive physics requires that children be able to explain their own reasoning about objects and forces). However, to the degree that children have an intuitive psychophysics, they should be able to represent the relative difficulty of discriminating perceptual evidence and these representations should guide their judgment and exploration.

And (lines 346-356) as follows:

The current studies also open up provocative questions for future research. They suggest that children have some metacognitive knowledge about their own ability to make perceptual discriminations. Anecdotally, some children also proffered explicit accounts of their own reasoning. In piloting Experiment 1 for instance, a child said that he preferred the more discriminable box because the pair was “more not the same”. Likewise, in Experiments 4-7, children sometimes explained their own reasoning (e.g., “this one’s gonna be hard”). Given the sophistication of the judgment required here (in which children had to compare observed data with unobserved alternatives), we believe children’s choices and exploration were less likely to underestimate their reasoning than asking children to justify their choices. However, further research might look at the extent to which children can explicitly account for the reasoning behind their decisions.

Another example: Experiments 1-3 are described as "Deciding what to explore". I agree (for me at least) that deciding what to explore is an important problem. The authors could make a careful case for how these experiments test abilities that the authors believe are important constituent abilities for choosing informative things to test or explore. The immediate tasks are not (to my mind) really exploration; they are (very interesting) decision problems that require some ability to reason (a la Raiffa's pre-posterior analyses) about exploration.

We appreciate the Reviewer’s feedback. We agree that since we (as the experimenters) explored both boxes and then asked children which box to open, Experiments 1-3 are better described as reasoning about exploration than as decisions about what to explore. We have changed the description per the Reviewer’s description (lines 94-96):

We report two series of experiments probing children’s intuitive psychophysics, considering first children’s reasoning about exploration, and second, their decisions about how long to explore.

An exploration variant (which I would also find interesting) could involve having the children select which of two boxes to shake, or to simply let them shake the boxes (and measure which

box is shaken first, how long or for what proportion of the time, etc). I don't think the authors need to do that experiment now, but they should be more careful in discussing what they did, as opposed to why they think it is interesting.

We agree with the Reviewer, and indeed, ran a version of the first suggested experiment, described in the SI as follows (lines 928-945):

Additional work

In addition to Experiments 1-3, we ran an additional study to see if children could infer the discriminability of the hypotheses without hearing the sound of the marbles shaken in the box at all. We used a method identical to Experiment 3 except that the experimenter never hid the box, put the marbles in the box, or shook the boxes; instead children were simply asked from the outset which pair of marbles they wanted to use for the box-shaking discrimination game, either a difficult to discriminate pair consisting of 8 and 6 marbles or an easy to discriminate pair consisting of 8 and 2 marbles.

In the first iteration of this experiment, 13 out of 16 children chose the unambiguous pair, but this effect did not replicate in a pre-registered additional sample of 24 children (15 children chose the unambiguous pair). Without any perceptual experience of the sounds of the marbles, it may have been difficult for children to reliably simulate the possible outcomes and the relative difficulty of the discriminations, or the simulations may have been too coarse to guide their explicit choice of which task to select. Alternatively, it's possible that after the simple warm-up task (Preliminary experiment, Object Identity), some children wanted a more challenging box-shaking game; they may have been sensitive to the difficulty of the discrimination, but, having not yet heard the sounds in the boxes, purposefully selected the harder game because it seemed more interesting.

The manuscript would already be quite great as a developmental paper, but for this journal I think it is important to connect to other strands of relevant literature (in science more broadly) and in the history of psychology. This is all very doable, and will broaden the range of readers who notice and appreciate the work here. More specifically:

Experiments 4-7 are very similar to the optimal stopping (or determining the size of the sample) experiments conducted by Wald and colleagues. For reviews see the 1967 Peterson and Beach Psych Bulletin paper on Man as an Intuitive Statistician. Craig McKenzie posted it at <http://pages.ucsd.edu/~mckenzie/Peterson&Beach1967PsychBulletin.pdf>; see page 37 on Determining the Size of the Sample. The earlier work was not (that I can remember) explicitly in a perceptual domain, but sets a nice foundation for the current work.

The intro nicely grounds the investigation in scientific inference, in particular the idea of discriminating among multiple hypotheses. It would be nice to cite at least a couple of classics in this area, like Chamberlin's (1897) Method of Multiple Working Hypotheses (<https://science.sciencemag.org/content/148/3671/754>), or Platt's (1964) Strong Inference (<https://science.sciencemag.org/content/146/3642/347>). There is also a lot of work in statistics and philosophy of science (Good, 1950; Lindley, 1956; Fedorov 1970s) that attempts to instantiate these ideas in a probabilistic Bayesian context, which would be relevant when thinking about the present Experiments 1-3. Beyond the developmental literature (which is nicely

cited) there is other recent work in psychology (the most famous being Oaksford and Chater's 1994 Selection Task model, <https://psycnet.apa.org/record/1995-08271-001>; Coenen et al.'s 2018 paper is a current overview/review, <https://doi.org/10.3758/s13423-018-1470-5>).

We agree these are valuable connections to draw and thank the reviewer for these suggestions. We have added these references to the most relevant parts of the text. In particular, we refer to Chamberlin (1897), Platt (1964), Lindley (1956), Good (1950) and Fedorov (2013) in the introduction (lines 49-52):

As in science, a rational learner should be able to estimate the sensitivity of her measurement apparatus (in this case, her perceptual system) to decide what would count as an informative experiment and amount of data given the alternative hypotheses she is trying to discriminate among (40-43).

We refer to Peterson & Beach (1967), Coenen et al (2018), and Oaksford & Chater (1994) two paragraphs later (lines 75-77):

Classic (44) and contemporary (45-46) work has examined the extent to which adult learning and exploration can be considered to be rational, before turning to related previous work on exploration in children, as in our original submission.

The article, at several points, dismisses the idea that children could be using simple heuristics to achieve their performance on the tasks. There is a great deal of work on people's use of simple heuristics (some of it even focusing on children), and recently on trying to build models that would make purely Bayesian (Marr's computational level) more psychologically plausible (bounded rational / Falk Lieder style). Why is none of this work (neither Gigerenzer-esque nor Griffiths/Lieder-esque) cited or considered in a serious way? It does not need to be in opposition to the more computational-style rationale for doing things in a particular way. As an example, the authors provide a very complex rationale for their model of perceptual discriminability from which d' is derived, as a basis for analyzing the results from Experiment 4-7. I trust this (and Josh McDermott is in the acknowledgements). I really wanted to know what the kids were doing. The authors must have some insight on this, after so many children did the task, videotaped, tested on an individual basis. I'm thinking of things like checking how far away from a 50:50 split the marbles are, and allocating observation time (shaking and listening) according to that. I would hope that a little bit of the authors' insight on this (it can be qualitative), after testing so many children, could be included in the article proper. Certainly it would be sensible to include a section in the Supplemental material on the authors' best, if tentative, ideas about the kinds of strategies that children used on the various tasks.

The Reviewer returns to this point later in the Review so we include that here:

Of great interest to test would be scenarios in which different possible heuristic strategies (my proportion-based heuristic, a heuristic that just looks at the difference in marble counts, etc.) contradict each other and/or the discriminability model.

We agree it is reasonable to consider more seriously alternative accounts based on simpler heuristics or resource-rational approximations to ideal Bayesian observers, and we now address this in the discussion. We explicitly test both of the heuristics the reviewer suggests as accounts of our Experiments 4-7, a numerical difference heuristic and a numerical ratio/proportion heuristic. However, we should note that our model is not intended merely as a computational-level account based on an ideal analysis of psychophysical discriminability. Rather we suggest a process-level account (although we do not model it in detail), whereby children mentally simulate the different possibilities that might be inside the box, as well as the sounds those objects would cause when the box is shaken, and can compare the imagined sound patterns to those they perceive when they actually shake the box based on what is actually in it. We take the measure of d' as a simple mathematical approximation to the discriminability children would judge from this process. The heuristics suggested by the reviewers seem to us to be not alternative process-level accounts of the same computation we are proposing, but rather alternative computations based only on the presented numerical information about the alternative hypotheses (e.g., 6 or 8 marbles), which might indeed be simpler in some sense than the more simulation-and-perception based account we favor. We now clarify this interpretation of our model, and at the same time address possible alternative heuristics and resource-rational accounts (lines 324-335):

Our account relies on mental simulation, and our quantitative results in Experiments 4-7 analyzed children's exploratory behavior using idealized models of perceptual discriminability in these mental simulations. However, it is possible that children might have relied on some simpler cognitive mechanism or heuristic (53), or a resource-constrained approximation to this ideal (54-55). One natural alternative to consider for Experiments 4-7 is that children took into account only a simple contrast in the linguistically and graphically presented number of marbles in each pair, without attending at all to the rich perceptual data they obtained in shaking the box or imagining possible sounds they might hear via mental simulations of box shaking. We evaluated two such heuristic models that avoid the computational burden that might accompany mental simulation, based on the absolute difference and (negative) ratio of the numbers of marbles in each pair. Both of these models perform well numerically (see SI, Additional Heuristic Models), and so it is indeed possible that children rely on such a mechanism in Experiments 4-7.

While we find that these heuristics do indeed represent credible alternative process-level accounts of our Experiments 4-7, we also articulate the reasons we favor the simulation-based account. First, it is more general, explaining our Experiments 1-3 as well as many other similar tasks children can perform that involve mentally simulated perceptual contrasts for a wide range of different objects and physical interactions not simply reducible to numerical ratios or differences among sets. Second, it appears more consistent with children's behavior in our studies; we follow the reviewer's suggestion above ("*I would hope that a little bit of the authors' insight on this (it can be qualitative), after testing so many children, could be included in the article proper*") to report some of our qualitative insights about what children appear to be doing in the task, in support of our preferred account. This argument comes directly after the text above (lines 336-345):

We believe, however, that mental simulation remains the best account of children's behavior. Experiments 1-3 demonstrated that children are able to reason about unheard objects that are neither marbles nor presented in sets of different cardinalities; the heuristics we evaluated do not apply in this domain (other heuristics, of course, might). By contrast, mental simulation offers a unified, and general, mechanism for performing all the experiments reported here as well as many other perceptual discrimination tasks. Another reason to prefer the mental simulation account stems from the heuristics' insensitivity to perceptual data; if children merely relied on heuristics, they would have no need to listen to the sounds of the box as they shook it but anecdotal observation suggests that children indeed listened closely to the sounds as they were exploring.

In addition to the above material inserted in the main text discussion, we have added a section to the supplemental materials explaining how we evaluated the two simple heuristic accounts described above (lines 1223-1265 of the supplement):

Additional Heuristic Models

We examined two potential heuristics that might underlie children's exploratory behavior. First, we considered whether a very simple cue, the difference between the number of marbles in each hypothesis (tube), could explain children's behavior. Formally we define the numerical difference heuristic as $nd = |l - m|$, where l and m are the number of marbles in a given contrast. nd is intuitively related to discriminability; a larger value indicates high discriminability, and a smaller value low discriminability (the exact relationship is unclear but we expect nd to increase monotonically with discriminability).

Second, we examined another alternative heuristic that takes the ratio of the larger to the smaller number of marbles as a predictor of exploration time. This heuristic formalizes the intuition of "distance from 50-50 split" – how far away a given pair is from having the same number of marbles in each set. Formally we define the numerical ratio heuristic as the ratio $nr = \frac{-l}{m}$, where l is the smaller and m is the larger number of marbles in a given contrast.

Both nd and nr are good quantitative predictors of children's box shaking time (Supplementary Figure S; nd : $r = 0.94$, 95% CI [0.76, 0.94], nr : $r = 0.95$, 95% CI [0.78, 0.95]). The fit of the nr heuristic is numerically indistinguishable from the d' measure we use; this should not be surprising as there is a close correspondence between the mathematical structure of these two measures, and they are themselves correlated at $r = 0.96$. The nd heuristic performs slightly worse, but there is a qualitative difference between its predictions and those of d' or nr . Across Experiments 4-7, there are four subsets of stimuli where the numerical difference is constant but discriminability d' and the numerical ratio nr differ, and intuitively the task seems more difficult when d' or nr are smaller: e.g., a numerical difference of 2 occurs with both contrasts of 4 v 2 marbles and 8 v 6 marbles, but 8 v 6 seems much more difficult than 4 v 2. This intuition is borne out by our empirical results. For contrasts scored equally by nd but not by d' , children on average explored more when the contrasts were less discriminable. Indeed, for each of the four numerical differences shared by more than one contrast, regression analysis revealed a

positive relationship between exploration time and negative discriminability (Fig. XX). Because each numerical difference corresponded only to at most four contrasts, none of these linear relationships is statistically significant on its own, but the overall pattern of a positive relationship in all four out of four possible subsets of contrasts is strongly suggestive of an effect of discriminability independent of absolute numerical difference.

Unlike *nd*, *nr* makes different predictions for different contrasts with the same numerical difference, in ways that are almost perfectly correlated with of *d'*. We therefore suggest that if a numerical heuristic turns out to provide the best explanation of children's box-shaking behavior – that is, if children were in fact explicitly estimating discriminability from the numbers of marbles shown rather than judging the discriminability of imagined perceptual evidence from alternative hypotheses via mental simulation – *nr* would be a more plausible heuristic account than *nd*. Because *nr* is so closely related to *d'* it might even serve as a resource-rational approximation of the ideal *d'*.

One small point is that the manuscript at several points notes that the data analysis was conducted by individuals who were blind to the experimental conditions. This is great. But isn't the coding of this (which box a child selected, how long a child shook a box) relatively objective? The bigger concern for me would be that due to the design of the experiment, the experimenter could not be blind to the experimental conditions. I don't think that should hold up the paper, but I think it would be helpful and appropriate to explicitly acknowledge this in the article, and perhaps to comment further in the Supplemental material on the rationale (or necessity) of the experimenter not being blind, why it should not matter, etc.

The Reviewer returned to this point later in the Review so we include that here as well:

I would like to see an acknowledgement of the fact that the experimenters were presumably not blind to the condition, and why the authors believe that that is not a problem. (Hans the horse effects).

We appreciate the feedback. It is the case that the experimenter knew the contrast but in none of the experiments was she in a position to easily influence the child's choice or exploration. In Experiments 1-3, the boxes were placed in a fixed location to the left and right of the child and she looked directly at the child when prompting the child to make a choice. In Experiments 4-7, the experimenter was sitting to the side of the child, out to the child's direct line of sight and did not interact with either the child or the box during the exploration period. Additionally, although the experimenter was not blind to the contents of the box, she was blind to the precise predictions across all 16 contrasts. Given the variability in each child's individual exploration time and the granularity of the distinctions (e.g., between 8 v. 6 and 6 v. 4) across the 16 contrasts, the experimenter was not in a position to influence the quantitative results. We now address this in the manuscript as follows:

(lines 145-148):

The experimenter was not blind to the contents of the box so to avoid her influencing the child's choice, the left/right positions of the box were fixed and the experimenter looked directly at the child during the prompt.

(lines 211-214):

The experimenter was not blind to the contents of the box but was blind to the precise predictions across all sixteen contrasts. She experimenter was positioned alongside the child, out of the child's direct line of sight and did not interact with the child or the box during the exploration period.

Other suggestions

Figure 1 and the description of the experiments in the body of the text were not adequate for me. The key point missing, which needs to be explicit in the figure, is that "each box contained exactly one item". I think this figure could be redone in a somewhat more helpful way. It can be done visually, but I illustrate how it might work schematically below:

Experiment 1. Box 1 contains a shiny pencil or a pillow. Box 2 contains a shiny pencil or a boring pencil.

Experiment 2. Box 1 contains a small elephant or a large elephant. Box 2 contains a small elephant or a small pig.

(etc.)

I think it would also be helpful to outline the basic experimental procedure in this figure or in a caption to this figure. Something like "Box 1 contains either the shiny pencil or a pillow. Box 2 contains either a shiny pencil or a boring pencil. Both boxes, when shaken, made sounds consistent with there being a pencil inside. You get to pick one box, open it, and keep the item that is inside it. Which box would you like to open?"

The Reviewer returned to this point later in the Review so we include that here as well:

Figure 1. It would help me if a pseudocode-style version of the experimental procedure were included here. In particular, it would be helpful to make clear that each box contained exactly one item. (Not everyone is working in this research area, and it does not go without saying that a box contains exactly one target.)

Thank you for the feedback. We have revised Figure 1 and the Figure caption according to your suggestions as follows (see also our revision of the description of the Method below):

(lines 155-159):

Figure 1. Schematic of Experiments 1-3 showing the more discriminable pair on the left and the less discriminable pair on the right (actual order counterbalanced). The leftmost item in each pair was the target. Only one item in each pair (the target) was placed in each box. Because the target was always placed in both boxes, the two boxes in each experiment made the same sound when shaken.

If children were told a little white lie, don't use euphemisms "children believed X but actually Y was true" to describe it. Just state what children were told and what was in fact the case.

The Reviewer returned to this concern later in the Review so we include this here:

[lines 373=375] "although children believed the tubes of marbles". Weren't the children told this? It is better to state what is known, namely what instructions the children were given. We don't (I think) know what the children believed. If there were some "white lies" along the way to make the experimental procedure work, it is really better to just state that clearly, because this kind of wording is hard to parse. Please check the manuscript for this; I think similar things came up a couple of other places (e.g. in the Supplementary Materials section).

We have revised both the manuscript and SI according to your suggestion (lines 431-434):

The tubes were pre-loaded with the appropriate number of marbles and sealed at the top; although children were told that the tubes of marbles would be poured into the box, marbles were in fact added quietly by hand to ensure that children did not get any evidence

about the sound until they themselves shook the box.

(lines 964-967):

Although children were told that the tubes of marbles would be poured into the box, marbles were in fact added quietly by hand to ensure that children did not get any evidence about the sound until they themselves shook the box.

Make sure that the paper is meaningful to people who are not developmental or cognitive psychologists. There are things like the years;months of age convention, which are not generally known, but are used without explanation in the manuscript.

We have now added this explanation to the manuscript (lines 110-111):

Throughout, we adopt the convention in developmental psychology of reporting children's ages as years;months (e.g., a mean age of four years and four months is written 4;4).

Test the method description (in the body of the manuscript) plus figure captions with someone who is in a different area (e.g. a biophysicist who does not do anything cognitive) to see if they can understand it. I found it hard, from the manuscript, to figure out what exactly was done. I think I did in the end, but it required a couple of iterations through the manuscript to do so; that should not be the case.

We apologize for the lack of clarity. We have revised the method description to spell out precisely what we did and we have asked readers not in the field to review the description to ensure that it is now clear (lines 130-144):

In Experiment 1 (see Fig. 1 and SI for details), children were introduced to two boxes. A pair of objects was placed in front of each box. Each pair consisted of an exciting target object (a pencil with a shiny holographic coating) and a boring distractor. The target was identical in both pairs. In the less discriminable pair, the distractor was an object that would make a very similar sound when shaken inside the box (a standard No. 2 pencil). In the more discriminable pair the distractor was an object that would make a very different sound when shaken inside the box (a small pillow). The experimenter pointed to the shiny pencil and the boring pencil and told the child, "I'm going to take just one object -- either the shiny pencil or the plain pencil -- and put it in this box here." Then she pointed to the other pair and the other box and said, "And then I'm going to take just one object -- either the shiny pencil or the cotton pillow -- and put it in this box here." She put up an opaque screen and removed all the objects from the child's line of sight. She silently put a shiny pencil in each box and then returned the boxes to the table. She told the child, "Remember inside this box, there could be either a cool shiny pencil or the plain yellow pencil"; "Remember, inside this box, there could be either a cool shiny pencil or the pillow"; (order and L/R position counterbalanced).

The abstract needs proofread and rethought. It should give a better idea of what was done and

why it is important.

Suggestions with respect to specific parts of the manuscript follow.

Abstract

If the authors (as I hope) are serious about the idea that testing among hypotheses should be the goal in science, it is funny to state that the value of evidence is "for testing a [single] hypothesis". The whole idea, starting at least with Chamberlin's (1890s) work, more recently with Good, Lindley, Platt, and others, is that scientists should figure out (and have as their goal) how to discriminate among many hypotheses, rather than simply to test a single hypothesis. I would delete "for testing a hypothesis" in the first sentence.

Please proofread the first sentence of the abstract. Either insert "that" before "are under" or delete "are".

"Children's exploration time was independent of the evidence heard". This is hard to understand, in the abstract. I think the sentence can be written in a clearer way, perhaps simply "Children's exploration time, across 16 contrasts, quantitatively tracked the discriminability of heard evidence from an unheard alternative".

We appreciate the feedback. We have rewritten the abstract, incorporating the Reviewer's suggestions, as follows (lines 12-22):

Abstract: Effective curiosity-driven learning requires recognizing that the value of evidence for testing hypotheses depends on what other hypotheses are under consideration. Do we intuitively represent the discriminability of hypotheses? Here we showed children alternative hypotheses for the contents of a box and then shook the box so children could hear the sound of the contents. Children were able to compare the evidence they heard with imagined evidence they did not hear but might have heard under alternative hypotheses. Across seven experiments, children (N = 160; mean: 5;4) preferred easier discriminations (Experiments 1-3) and explored longer given harder ones (Experiments 4-7). Children's exploration time, across 16 contrasts, quantitatively tracked the discriminability of heard evidence from an unheard alternative. The results are consistent with the idea that children have an *intuitive psychophysics*: children represent their own perceptual abilities and explore longer when hypotheses are harder to distinguish.

Intro

[line 54] "subjective discriminability of competing hypotheses". The word "subjective" here put me off on the first read, and I actually think it is not consistent with the experiments conducted, or at best is confusing. My understanding of the d' and other discriminability models used in Experiments 4-7 is that they are intended to be objective models of how hard particular discriminations are for people to make. "Subjective", to me, refers to people's (in this case children's) understanding of how difficult a particular discrimination will be.

We have eliminated the word “subjective” throughout.

[lines 75-75] I'm not sure what "looked only at qualitative relationships between children's uncertainty and exploration" means. Certainly these studies quantified the things they were measuring, and manipulated uncertainty in various ways.

Previous studies tested whether there was an effect of uncertainty on exploratory behavior but no previous work has attempted to predict *how much* of an effect there would be (with, as we note, the exception of Kidd, Aslin, & Piantadosi’s work showing a U-shaped relationship between infant looking-time and the predictability of events). That is, previous work has manipulated uncertainty and provided statistical analyses of the qualitative effects (i.e., finding that children explore significantly more in Condition A than Condition B). But previous work has not tried to predict or measure how much a change in uncertainty leads to a change in exploration. By contrast, here we predict and find that quantitative increases in the difficulty of discrimination contrasts lead to quantitative increases in children’s exploration.

[lines 200-201]. The proportional playtime statistic seems to be a reasonable dependent variable to measure. It is helpful to note that results also hold if using log playtime. Did the authors also test using raw playtime?

As the Reviewer notes, the results hold for both proportional and untransformed playtime reported in log seconds but raw playtime was not normally distributed, violating the assumptions of our statistical tests, so we did not run the analyses using raw playtime. We now highlight this in the. SI as follows (lines 1053-1055):

The children’s raw playtime was not normally distributed, violating the assumptions of our statistical tests so we only considered inferential statistics on log-transformed playtime (which is normally distributed).

Figure 5. Please check the within-figure headings: I think Experiment 5 through 8 should be numbered Experiment 4 through 7 to match the numbering elsewhere in the paper.

Thank you; we have corrected the error.

References: A nice set of reference. It would be helpful to explicitly connect to some older and other strands of relevant work.

We appreciate the reviewer’s earlier suggestions for references to several lines of older work, as well as more recent relevant work, and we have included many of these references in the text as described above.

Supplemental Materials: It was hard to get to know the experimental methodology. Referring explicitly back to the figures in the main text, where appropriate, when describing experiments in the Supplemental materials, would be helpful. Figures 1 and 2, as previously noted, could be expanded to provide some more helpful information. If space constraints preclude doing this in

the main text (which I doubt-- I think you can do it), then comprehensive figures to describe the procedure should be included in the Supplemental materials.

Per the Reviewer's suggestions, we have revised Figure 1 and 2 and our description of the methods, as detailed above. We now also refer back to these Figures in the SI at the end of the Procedure section for each of Experiments 1-3 (line 794, and line 834, and line 918):

See Figure 1, main text.

and in the Procedure section for Experiments 4-7 (line 1017):

See Figure 2, main text.

[line 578] do you mean to italicize Volume Control, to be consistent with naming of other preliminary experiments?

Thank you. We have fixed the error.

[lines 832-840] It is important to state, in numbers, what happened with the "pre-registered additional sample of 24 children". It can leave the reader with a bad impression if you decline to provide explicit numbers in cases where the numbers are unhelpful. Just state what happened.

Thank you for the feedback. We now report the results in detail as follows (lines 936-938):

In the first iteration of this experiment, 13 out of 16 children chose the unambiguous pair, but this effect did not replicate in a pre-registered additional sample of 24 children (15 children chose the unambiguous pair).

We very much appreciate your and the Reviewer's thoughtful feedback. Please don't hesitate to let us know if there is more information we could provide.

Sincerely,

Max Siegel, Rachel Magid, Madeline Pelz, Josh Tenenbaum, and Laura Schulz

***REVIEWERS' COMMENTS:

Reviewer #1 (Remarks to the Author):

The authors have thoughtfully and thoroughly responded to my questions and concerns.

Reviewer #2 (Remarks to the Author):

review of revision 1 of Siegel, Magid, et al.

The revised manuscript is much clearer in description of the experiment methods; at least attempts to address possible concerns and alternate hypotheses; is more explicit about the differences between the authors' interpretations and the data themselves; connects to a wider variety of strands of historical and current literature; and presents the obtained data in a richer way (now also including bits of insightful quotes from the children doing the tasks). Although each of the individual changes were fairly minor, the net result in my view is a manuscript that is close to ready for publication to a broad scientific audience.

I have some remaining concerns that I hope the authors will take into account in the next version. From my standpoint, if the authors do so and explain what they have done in the next cover letter, I do not need to review the next version; I will trust them on the details.

-Jonathan Nelson

moderate scientific points

I think that the first part of the manuscript, up through discussion of experiment 3, will still unnecessarily alienate readers from the simple heuristics tradition. There is no need to do that. Why reduce the size of your audience, and your impact, unnecessarily? You have a cool set of experiments and results. I suggest to go through the first parts of the paper and carefully look at the wording on this. For experiments 4-7, the authors tested out some simple heuristics (at least one of which I suggested, off the top of my head, in the first iteration of the manuscript) and found that they are quantitatively indistinguishable from the more sophisticated perceptual simulation in how well they match the obtained data. For experiments 1-3, there was no implementation of simple heuristics, yet there are statements like "rather than relying on simple heuristics", which I personally find to be off-putting. As a heuristic I would suggest to figure out what sensory modality is most relevant for the discrimination task (e.g., what you could perceive without looking at the object, something about sound or weight) and checking which box best discriminates on that sensory modality. This would not require any complicated simulation architecture. The authors can write the paper, to some extent, as they wish, but why alienate parts of your audience by speaking out against ideas that either (1) you did not test, or (2) that you consider in only the most straw-man form? This also applies to the points in the discussion, claiming "if

children merely relied on heuristics, they would have no need to listen...". That claim is true if you take the authors' most straw man version of possible heuristic strategies, but emphasizing this thing tends to reduce the authors' apparent credibility.

"learners' tendency to explore more when the probability of information gain is higher". Most of the cited papers are talking about the expected information gain, not a probability of information, as opposed to zero information. This could be reworded in a more accurate way.

typos and wording things

it would be sensible to state once how the confidence intervals were derived (there are many ways to do this for a proportion)

typo in the abstract: a "form" should be "from"

although the authors now explain the years;months convention for reporting children's ages in developmental research, the abstract comes before this explanation. Abstracts are supposed to be self-contained for anyone who is in the desired audience, in this case meaning a broad scientific audience. In the abstract it would be better to spell out "mean 5 years, 4 months".

the intro does a better job now of explaining what happened in the experiments. Figure 1 is now nice in how it explains the procedure. Cite in in the second sentence of the "We report two series of experiments" paragraph in the intro, so that the readers can immediately refer to it.

typo: "She experimenter was positioned"

there is a five-line sentence that addresses heuristic models for experiments 4-7, variant psychophysical models, etc.; at the end of that sentence, a key finding, namely "produce nearly identical results" is buried. Make that result its own sentence, at least; otherwise it looks like you are trying to bury it.

We have made the specific changes requested by Reviewer 2:

I think that the first part of the manuscript, up through discussion of experiment 3, will still unnecessarily alienate readers from the simple heuristics tradition. There is no need to do that. Why reduce the size of your audience, and your impact, unnecessarily? You have a cool set of experiments and results. I suggest to go through the first parts of the paper and carefully look at the wording on this. For experiments 4-7, the authors tested out some simple heuristics (at least one of which I suggested, off the top of my head, in the first iteration of the manuscript) and found that they are quantitatively indistinguishable from the more sophisticated perceptual simulation in how well they match the obtained data. For experiments 1-3, there was no implementation of simple heuristics, yet there are statements like "rather than relying on simple heuristics", which I personally find to be off-putting. As a heuristic I would suggest to figure out what sensory modality is most relevant for the discrimination task (e.g., what you could perceive without looking at the object, something about sound or weight) and checking which box best discriminates on that sensory modality. This would not require any complicated simulation architecture. The authors can write the paper, to some extent, as they wish, but why alienate parts of your audience by speaking out against ideas that either (1) you did not test, or (2) that you consider in only the most straw-man form? This also applies to the points in the discussion, claiming "if children merely relied on heuristics, they would have no need to listen...". That claim is true if you take the authors' most straw man version of possible heuristic strategies, but emphasizing this thing tends to reduce the authors' apparent credibility.

We have eliminated references to simple heuristic accounts in the framing of Experiments 1-3, and the Discussion. As the reviewer indicates, our studies (especially Exps 1-3) are not primarily intended to test these accounts, and we agree that it would be better to avoid unnecessarily alienating readers from that tradition.

"learners' tendency to explore more when the probability of information gain is higher". Most of the cited papers are talking about the expected information gain, not a probability of information, as opposed to zero information. This could be reworded in a more accurate way.

We have corrected the phrasing in our discussion of previous work to refer to studies exploring when "expected information gain" is higher, rather than "probability of information gain" as in the original version. We agree with the reviewer that our previous phrasing was ambiguous and the new terminology is clearly better.

We have also corrected the typos and made the minor writing improvements that the reviewer identified:

it would be sensible to state once how the confidence intervals were derived (there are many ways to do this for a proportion)

We now specify how the confidence intervals are derived for Experiment 1 (page 3):

95% CI [0.67-1], Bernoulli normal approximation CI.

although the authors now explain the years;months convention for reporting children's ages in developmental research, the abstract comes before this explanation. Abstracts are supposed to be self-contained for anyone who is in the desired audience, in this case meaning a broad scientific audience. In the abstract it would be better to spell out "mean 5 years, 4 months".

We agree and have made the suggested change.

the intro does a better job now of explaining what happened in the experiments. Figure 1 is now nice in how it explains the procedure. Cite in in the second sentence of the "We report two series of experiments" paragraph in the intro, so that the readers can immediately refer to it.

We now refer the reader to the illustrative figure (page 2):

In Experiments 1-3, an experimenter shakes two boxes, generating identical sounds (see Fig. 1).

there is a five-line sentence that addresses heuristic models for experiments 4-7, variant psychophysical models, etc.; at the end of that sentence, a key finding, namely "produce nearly identical results" is buried. Make that result its own sentence, at least; otherwise it looks like you are trying to bury it.

We have extracted the finding into a new sentence (page 8):

These produce nearly identical results for our purposes.